# Zinc-Based Metal-Organic Frameworks in Drug Delivery, Cell Imaging, and Sensing

**DOI:** 10.3390/molecules27010100

**Published:** 2021-12-24

**Authors:** Rashda Safdar Ali, Hongmin Meng, Zhaohui Li

**Affiliations:** Key Laboratory of Functional Nanomaterial and Medical Theranostics, Henan Joint International Research Laboratory of Green Construction of Functional Molecules and Their Bioanalytical Applications, College of Chemistry, Institute of Analytical Chemistry for Life Science, Zhengzhou University, Zhengzhou 450001, China; Rashda-safdar@hotmail.com (R.S.A.); hmmeng2017@zzu.edu.cn (H.M.)

**Keywords:** Zn-MOFs, drug delivery, cell imaging, sensors

## Abstract

The design and structural frameworks for targeted drug delivery of medicinal compounds and improved cell imaging have been developed with several advantages. However, metal-organic frameworks (MOFs) are supplemented tremendously for medical uses with efficient efficacy. These MOFs are considered as an absolutely new class of porous materials, extensively used in drug delivery systems, cell imaging, and detecting the analytes, especially for cancer biomarkers, due to their excellent biocompatibility, easy functionalization, high storage capacity, and excellent biodegradability. While Zn-metal centers in MOFs have been found by enhanced efficient detection and improved drug delivery, these Zn-based MOFs have appeared to be safe as elucidated by different cytotoxicity assays for targeted drug delivery. On the other hand, the MOF-based heterogeneous catalyst is durable and can regenerate multiple times without losing activity. Therefore, as functional carriers for drug delivery, cell imaging, and chemosensory, MOFs’ chemical composition and flexible porous structure allowed engineering to improve their medical formulation and functionality. This review summarizes the methodology for fabricating ultrasensitive and selective Zn-MOF-based sensors, as well as their application in early cancer diagnosis and therapy. This review also offers a systematic approach to understanding the development of MOFs as efficient drug carriers and provides new insights on their applications and limitations in utility with possible solutions.

## 1. Introduction

Metal-organic frameworks (MOFs) are fascinating hybrid porous materials formed by a combination of metal and organic compounds [1,2]. The coordinated organic ligand serves as linkers between varied metal interiors [3]. Kinoshita and colleagues published the first report on metal-organic frameworks in 1959. Later, in the 1990s, Hoskin and Robson raised interest in this class of materials by adopting further logical methods to its “reticular” layout and synthesis [4]. These hybrid MOFs are found in one-dimensional, two-dimensional, or three-dimensional structures [5]. The primary objective for the development of MOFs was to achieve the desired characteristics by tuning their composition via changing either the metal or the organic component compared to their individual components [6]. MOF cavities permit the encapsulation of different guest molecules to allow drug release and optimal imaging [7,8]. If the volume of MOFs particles has been reduced to the nanoscale, nano-MOFs (NMOFs) could function as effective nano carriers to supply agents for imaging, chemotherapy, photo thermal therapy, photodynamic treatment, and rehabilitation [9].

There are certain properties associated with MOFs, such as their versatile architecture, increased internal surface area, and ease of tuning their configuration by altering any of their components, i.e., either metal or organic ligand, that make them ideal candidates for applications in drug delivery systems, imaging, and sensors [10]. Therefore, the selection of components in MOFs should be such that it is environmentally and biologically compatible [11].

It is imperative to ensure the non-toxicity of MOFs for their successful application in biological domains such as intracellular imaging and drug carriers [12,13,14]. In such instances, an important factor in designing MOFs is the likelihood to create mixtures of bioactive cations (Ca^2+^, Zn^2+^, Ag^+^, and Fe^2+^/^3+^) with bioactive molecules as organic linkers that can medicinally transfer the component of interest through degradation of the framework [15]. MOFs appeared to possess various unique characteristics over conventional nanocarriers such that they can be easily undergo modification (in situ synthetic or post-synthetic modification) [16,17], which have made them ideal compared to conventional nanocarriers [18].

In recent years, many review studies have been published on MOFs and its applications [19,20,21]. Modifications in design and type of organic ligand (such as phosphonates, imidazolates and carboxylates, etc.) and metal (i.e., Ca, Fe, Ag, and Zn, etc.) result in the fabrication of such MOFs that have utmost tunable porosities, rendering them perfect choices for holding, storage, and formation of effective drug delivery structures; and encapsulation of drugs, active gases, cosmetics, imaging, and effective theranostics [22,23,24,25,26,27]. However, to our knowledge, no review has been published, especially on the developments of Zn-MOFs nano carriers for drug delivery, cell imaging, and sensing applications, as summarized in Figure 1. Zinc, as an endogenous low-toxic transition metal cation, has become an ideal candidate in the preparation of zinc-based MOFs for applications in the biological system particularly as drug carriers [28]. In addition, zinc has been extensively used in dermatology a skin moisturizing with astringent, antibacterial, and anti-inflammatory properties [29,30].

The broad inner surface area and the flexible architecture of MOFs make them successful competitors for the production of developed drug delivery systems, sensing, and imaging [31,32]. Hydrophilic/hydrophobic interactions may be demonstrated by MOFs, and even a versatile arrangement that allows their porosity to be modified relative to physical and chemical properties for various drugs or gas can be obtained [33,34]. In most biographies, for example, biological processes, such as systems of drug delivery or nontoxic MOFs, were certainly preferred rather than toxic ones in intracellular imaging. The chemical stability of zinc ions as the desired component is recognized in bioapplication domains for implementing Zn-based MOFs, particularly as carriers of drugs [35,36,37,38]. Therefore, MOFs must be recognized for various applications, structured from naturally and chemically mediated essential elements that are competitive with the environment. However, in most research, non-toxic MOFs were preferred in intracellular imaging than toxic ones [39,40].

## 2. Synthesis of Zn-Based MOFs

Mechano-chemical, hydrothermal, microwave-assisted, and ultrasound procedures have been adopted to develop MOFs with practical attributes and desired texture [41,42,43]. The prime benefit of creating MOFs is the possibility of tuning their structure and functionality during their fabrication, where guest compounds bind and react with open sites. The selection of suitable building blocks during reticular synthesis ensures the creation of predefined shapes cavities. Using these approaches to make MOFs allows you to tune their structure and operation throughout the fabrication process, where visitor compounds can connect to vacant spaces and react with them [44,45]. Various iso-reticular MOFs formulations have been prepared with zinc-based octahedral clusters (Zn_4_O(CO)_2_) serving the metallic centers surrounded by varied organic dicarboxylate nexus, thus constituting contrary 3D cubic structures [46,47]. The free and fixed pores diameters range at 3.8–19.1 Å and 12.8–28.8 Å in isoreticular MOF-1 (IRMOF-1) and isoreticular MOF-6 (IRMOF-16), respectively [48,49]. However, altering the organic linkers in MOFs variates the pore dimensions of the resulting framework and creates those with variable affinities for guest compounds [50]. For example, as the organic linkers are altered in IRMOFs, selective hydrogen adsorption and diffusion in the porous structure are varied [51].

### 2.1. Hydro or Solvo/Thermal Technique

Previously, the hydrothermal technique was used for producing coordinated metal-organic composites [52,53]. The problem encountered during the formation of these composites was the exclusion of remaining solvent from the matrix of MOFs caused by external heating. On the other hand, high temperatures could risk the composition of developed MOFs, which can be cut down by low-temperature solvent removal [54,55]. However, Zn-based MOFs that have been synthesized with stable thermal and chemical properties were reported [56]. In particular, hydrothermal, microwave-hydrothermal, and microwave-solvothermal methods are very low-temperature methods for processing nanophase materials of various sizes and shapes. These procedures conserve energy and are environmentally sustainable since the reactions take place in discrete, closed system environments. Using the above methods, the nanophase materials can be produced in either a batch or continuous process. This takes a long time (typically half to several days) and requires high energy (over a thousand watts) relative to the traditional hydro/solvothermal heating system [57]. One set of MOFs was released in 2013. 1-dinebsional chains of tetrahedral ZnO_2_(OH)_2_ and octahedral ZnO_2_(OH)_2_ combine together by a 3-hydroxy bridging agent and produces Zn_2_(BTC)(OH) (H_2_O) 1.67H_2_O. The carboxylate groups of BTC moieties also connect octahedral and tetrahedral entities. The [Zn_2_O_6_(OH) (H_2_O)]n chains are linked in this manner to form an expanded three-dimensional structure [58].

### 2.2. Electrochemical Method

For the first time, the mixed-ligand MOF [Zn(1,3-bdc)0.5(bzim)] was effectively synthesized electrochemically in this study. It discovered that the amount of current and reaction time were the important elements influencing quality and yield. The authors were able to synthesize pure desirable MOF (crystallite size of 32.3 nm) with a substantially greater products (87%) compared to previously reported methods such as hydrothermal, solvothermal or slow diffusion techniques. Furthermore, reaction time was reduced from at least 8 days (solvothermal approach) to only 2 h. According to early UV-Vis measurements, the material absorbed a large quantity of Ibuprofen (163.9 mg/g), which was comparable with the result found by ^13^C-NMR spectroscopy (160.7 mg/g) [59].

### 2.3. Ultrasound Methods

Environmentally friendly and budget friendly nanoscale MOFs have been synthesized by developed ultrasound methods [60]. The ultrasonic technique is more effective in fabricating Zn-based MOFs with fluorescent characteristics, mainly for the accurate sensing of ethylene amine. Similarly, as demonstrated in Figure 2, varying the sonication time can result in the different morphologies of Zn-based MOFs, i.e., nano sheets, nano belts, and microcrystals [61]. An example is [[Co_2_(TATAB)(OH) (H_2_O)2]·H_2_O·0.6O]. In order to create nanostructures, the experiment was first constructed in an autoclave for 72 h at 180 °C and then irradiated for 15 min with ultrasound (80 W). Ultrasounds, on the other hand, have reduced the response time to 2 h [62]. In comparison to previous approaches, ultrasonic irradiation reduces response time while simultaneously improving particle size and shape. [Zn_2_(NH_2_-BDC)2(4-bpdh)] is a Zn(II)-based MOF. 3,4-bis(4-pyridyl)-3,4-diaza-2,4-hexadiene, DMF=N,N dimethylformamide) is made (NH_2_-BDC=amino-1,4benzenedicarboxylate,4-bpdh=2,5-bis(4-pyridyl)-3,4-diaza-2,4-hexadiene, DMF=N,N dimethylformamide). The direct pyrolysis of S-TMU-16 NH_2_ MOF was used to produce ZnO octahedral nanostructures in a simple and easy manner [63]. Two 3D porous Zn-based MOF prepared by the sonochemical process is [Zn(oba)(4-bpdh)0.5]n·(DMF)1.5 (TMU-5) and the other one is [Zn(oba)(4-bpmb)0.5]n (DMF)1.5 (TMU-6). Further calcination of these MOFs at 550 °C enhances RhB adsorption. The conclusion is that by changing the sonication time, we can achieve MOFs with different morphologies, sizes, and shapes via ultrasound-assisted irradiation methods [64,65]. [Zn(NH_2_-BDC)(4,4′-bpy) is a 1.7-micrometer-sized Zn-based MOF synthesized in 60 min by ultrasound method. Moreover, this combination of Al^3+^ and NH_2_BDC ligand induces an excellent potential as the fluorescent sensor for low limit detection of toxic Al^3+^ and methanol in MOFs [66].

Under ultrasonic irradiation, several chemical processes have been performed in high yield with reduced reaction durations [67]. The interaction of Zn(OAc)_2_ and H_2_BDC in DMF during ultrasonic irradiation for 10 min yielded nanostructures with a breadth of 150–300 nm and a height of 2–5 µm. Whenever the response time was too long at 20 min, nanoparticles became larger in size. Having diameters ranging from 300 nm and length ranges 2 µm, such nanomaterials have regular quadrate morphologies. Improvements in reaction time were reflected in a 1.5% boost in nanostructures dimensions. However, when the reaction time was prolonged beyond 60 min, numerous microcrystals with size varying from hundred nanometers to tens of nanometers were obtained. Once reaction length increased to 90 min, massive microcrystals could not be identified, although nanoparticles of various shapes and sizes and those that were irregularly shaped were observed. This result indicated that throughout long-term ultrasonic irradiation, Zn(BDC) (H_2_O) n microcrystals were degraded, and all these microcrystals were transformed into nanoparticles caused by weak covalent bonding among 2D conjugated polymers.

### 2.4. Microwave

The microwave-assisted method is another budget-friendly technique for developing zinc-based MOFs that supplements benefits including improved reaction rate, practical assessment of process parameters, and control over facile morphology [68]. Meanwhile, this method is considered more suitable due to the synthesis of MOFs from novel material and narrow particle size distribution [69]. Microwave-assisted chemical precipitation is an excellent approach for incorporating b-estradiol. Moreover, the nano-sized crystals of Cr-MIL-101 were obtained by microwave heating at 210 °C in 2011 [70].

However, Sabouni with his team first recorded microwave-assisted production of CPM-5 in 2012 [71]. Microwave-assisted heating is a greener technology for synthesizing materials in a quicker timescale (a few minutes to hours) and with lower power consumption due to direct and uniform heating (hundreds of watts). Figure 3 illustrates how well the crystal shape of MMOF-5 changes as illumination duration advances. MMOF-5 crystal faces showed smooth and shiny during 15 to 30 min; however, as illumination frequency increased, interface imperfections developed. The polymeric binder inside the precipitated MMOF-5 appears to dissolve between 45 and 60 min, producing instability [72].

### 2.5. Templated Synthesis

Because of their capacity to regulate crystal growth, form, and secondary porosity, template-directed approaches for the synthesis of MOFs have opened up new areas in materials research. The template was utilized to direct the evolution of MOFs toward better nano-architectures with enhanced functions. In general, hard templates for creating the nanoscale MOFs, including semi-sacrificial, sacrificial, and non-sacrificial hard templates, have been provided in the literature. Due to the fact that considerable template-based synthesis research has been published in a number of recent publications, we will discuss a few instances of template-based synthesis below.

Hwang and colleagues presented a simple but productive synthesis process for MOFs in regulated polymorphism, crystalline shape, and size that could be carried out under ambient air conditions. The influence of solvents on ZBD synthesis was explored firstly in a meaningful manner in this study by varying the kind and concentration of the solvents. The solvent template effect allows for the synthesis of one-dimensional rod-like ZBDh in DMF and two-dimensional plate-like ZBDt in MeOH. The control of crystallite size and aspect ratios was performed by combining DMF/MeOH in different ratios such as a co-solvent. Each solvent acts as a crystal modulator, slowing nucleation and increasing crystal size; for example, increasing the concentration of MeOH increases the crystal size of ZBDh. The architecture shifts from 1-dimensional rods to 2-dimensional plates when MeOH is added. As a proof of concept, the resulting ZBDs were dynamically transformed into hexagonal rod-nanoporous and tetragonal plate-nanoporous carbons for usage in EDLCs. The produced ZBDs were cylindrically polymerized to hexagon shaped rod and tetragonal plate nanoporous carbons and used as energy storage devices to help in finding basic morphology–volumetric capacitance correlations in supercapacitors [73].

Hou’s group reported a self-templated technique for constructing a hydrangea-like framework of open carbon cages using morphology-controlled thermal alteration of core@shell MOFs. Direct polymerization of core@shell zinc (Zn)@cobalt (Co)-MOFs yields open wall nitrogen-doped carbon cages with well-defined walls. Upon the introduction of guest iron (Fe) ions into the core@shell MOF precursor, the open carbon cages self-assemble to construct a hydrangea-like 3D structural frame. This system is interconnected by carbon nanotubes, which are expanded in situ on Fe–Co alloy nanoparticles constructed during the pyrolysis of Fe-introduced Zn@Co-MOFs. This structural frame of open carbon cages displays effective bifunctional catalysis for both OER reactions, achieving great performance in Zn–air batteries [74]. Li and colleagues utilized PS nanospheres with a diameter of 400 nm as a hard template to create hollow spherical ZIF-8 by spreading the ZIF-8 precursors (Zn^2+^ and 2-methylimidazole (2-MeIm) and constructing a ZIF-8 shell on the surface of the PS nanospheres. The removal of PS from the framework by toluene treatment resulted in the creation of a hollow spherical ZIF-8. When ZIF-8 is manufactured without a template, the protein takes on a micron-sized dodecahedral structure with a non-hollow architecture. In the cycloaddition process of 1,3-cyclohexanedione, hollow spherical ZIF-8 materials and unsaturated aldehydes displayed a better conversion rate of 89 percent over extensively biosynthesized ZIF-8 material (73 percent). This improvement is feasible due to an increase in accessible surface area and, as a consequence, an improved active catalytic site for the cycloaddition process [75,76].

Wang and colleagues devised a unique method for developing bimetallic Zn–Co ZIFs by employing ZnCo-LDH as a sacrificial template (merged ZIF-8 and ZIF-67). In their approach, ZnCo-LDH was first dispersed in a mixed solution of DMF and water (DMF:water = 10:1), followed by the addition of a preset quantity of 2-MeIm. This technique led the Zn–Co LDH to lose its identity and it transformed into a two-dimensional MOF structure. During the first four hours of the reaction period, ZnCo-ZIF nanostructures with a diameter of 100 nm started nucleating on the ZnCo-LDH’s interlayer surface. Nonetheless, during the following 4 h, packed ZnCo-ZIF polyhedra with compact flake-like shape formed, and 2-MeIm completely sacrificed LDH and reassembled into Zn–Co ZIF. Following pyrolysis, this extensively intercalated ZnCo-ZIF formed a remarkable honeycomb-like macroporous structure with adjustable electrochemical and supercapacitance capabilities [77].

In a groundbreaking study, El-Hankari and colleagues used three distinct nanoshapes of ZnO (e.g., nanosphere, nanorod, and nanostar) coated with dodecyl guanidinium hydrochloride (C12Gua) as an etchable and sacrificial hard template for the growth of morphologically tunable ZIF-8 via a simple surface-passivation method. The reaction conditions carefully controlled the formation and development of ZIF-8 nanostructures (e.g., emulsifier type and amount, temperature, solvent, and time). It was also crucial to employ a suitable amount of C12Gua surfactant while controlling ZIF-8 development on ZnO nanostructure. One benefit of this method is the use of aqueous systems rather than volatile chemicals (such as MeOH and DMF) for etching the ZnO template. El-Hankari and colleagues used a PVP-assisted base etching approach to selectively remove the amphoteric ZnO core while preserving the ZIF-8 shell. Furthermore, since ZIF-8 was retained after etching, adding Au NPs to the C12Gua-coated ZnO core had no influence on the etching process. As a consequence, the Au NPs inside the ZIF-8 shell formed a hollow Au-ZIF-8 yolk–shell architecture. The authors also discovered that ZnO-ZIF-8 is a rhodamine B adsorbent. As a result, ZnO-ZIF-8 outperformed ZnO (5.8 percent) and ZIF-8 (38 percent) in terms of adsorption efficiency [78].

Won-Tae Koo and colleagues created nanosized Hollow Spheres of Catalyst-Loaded ZnO/ZnCo_2_O_4_. In this study, the templating procedure using PS sphere and BM-MOF was employed to effectively functionalize ultrasmall Pd NPs on TMO HSs. The theorized Pd-ZnO/ZnCo_2_O_4_ HSs have a high surface area and gas permeability due to their hollow design. By purposely breaking the ZnO phase, heterojunctions are also generated in the Pd-ZnO/ZnCo_2_O_4_ HSs. Furthermore, because of their small size and homogenous distribution, the NPs are an excellent catalyst for the complex production of ZnO/ZnCo_2_O_4_ HSs. Pd-ZnO/ZnCo_2_O_4_ HSs produced demonstrated extraordinarily strong selectivity toward acetone gas with remarkable sensitivity (*S* = 69 percent to 5 ppm at 250 °C) [79]. In addition, a list showing MOFs with their building blocks is shown in Table 1 [80,81,82,83,84,85].

## 3. Influence of Composition

Biologically friendly composition of metal-organic frameworks has enabled their application in biomedical and environmental fields; however, different parameters, including their use, associated risks and benefits, distribution, degradation, accumulation in various organs and tissues, and excretion from the body, are set to decide one composite over the other [1,18]. Therefore, the selection of different metals and organic linkers at different rates depends on the aforementioned conditions [86]. On the other hand, the most suitable options for the inorganic part of composites include calcium, zinc, magnesium, iron, and zirconium, and the selection of specific metal relies on its lethal toxicity [87]. The other advantage of organic linkers is their functionalization, providing the option of creating a combination of different types of organic functionalities, thus altering the physical and chemical properties of the solid. An earlier study, for example, documented the synthesis of certain MOF composites in the presence of polyoxometalates, sulfonates, amines, phenolates, and imidazolates. Another benefit of organic linkers is their functionalization, which allows for the creation of either all-through or after-the-fact combinations of various kinds of organic functionalities (i.e., apolar and polar), hence modifying the physical and chemical properties of the solid. In the presence of POMs, Liu and colleagues synthesized some MOF composites (the production procedure was quite similar to the BAS technique), dubbed NENU-n, from in situ synthesis of MOFs (HKUST-1) [88]. Ishida et al. employed a solid grinding process to make Au-MOF composites by grinding Au particles with MIL-53(Al), MOF-5, and HKUST-11 [89]. As a result, MOF composites, in which MOFs are combined with a variety of useful materials, have been created to compensate for the flaws of individual components. MOFs can be combined with zero-dimensional materials such as nanoparticles and quantum dots to create new materials, one-dimensional materials such as nanorods, nano tubes, and nanobelts, two-dimensional nanosheet materials, and three-dimensional MOF composites such core-shell and cubic composites. In this review, we address recent breakthroughs in MOF and their advanced synthetic methods as well as their various applications, especially related to drug delivery, cell imaging, chemo sensing, and also some electrochemical applications such as supercapacitors, sensors, catalysts, and also batteries, based on their measurements. With yields of up to 99 percent and a catalyst loading of 3 mol percent, the chemical displayed significant activity as a heterogeneous catalyst, notably with cyclic secondary amines and saturated aliphatic aldehydes 41 [90].

Exogenous linkers are either synthetic or ones orignating from natural origins that would not interfere in normal functions in body. Exogenous organic linkers used in MOFs are made from zinc named as zinc adeninate-4,4′ biphenyl dicarboxylate BioMOF-1. Another option for selection of organic linkers is the one that are a constructive part of body composition. These linkers are ideal in term of their application in living cells, because of their least possible adverse effects when administered in the body. Several metal organic frameworks with such organic linkers have been developed so far. An example of MOF based on endogenous linker is a porous zinc amino acid metal organic framework that depicts a flexible framework having potential to accommodate carbon dioxide [91].

### 3.1. pH-Responsive Zn-MOFs Nanocarrier

Another achievement in the use of MOFs based on Zn metal for medical field uses is the design of pH-dependent medicinal carriers. Polyacrylic acid@zeolitic imidazolate framework-8 (commonly known as PAA@ZIF-8) possesses an improved doxorubicin (DOX) loading capacity of around 1.9 *g*/*g*, which is an anticancer drug. This is attributed to the electrostatic reactions between the positively and negatively charged DOX and PAA, respectively, and coordination interaction among Zn^2+^–doxorubicin. It has been found by using confocal laser scanning microscopy analysis that drug-loaded MOF releases quickly at a pH value of 5.5 in a buffer solution rather than at a neutral pH of around 7.4 [92].

One of the medication mechanisms of distribution of 5-FU@ZIF-8 (5-FU = 5-fluorouracil) has been reported recently [93]. The medication delivery system does have a 66% therapeutic efficacy and pH-responsive drug dissolution technology. Inside drug dissolution research, 5-FU had just been launched quickly at pH 5.0 as compared to those at acid levels 7.4, with rather upwards of 45% of both the drug expelled between one hour at acidic pH, particularly in contrast with only 17% at pH 7.4. The research has always used pH-responsive NMOFs in DDS for drug delivery applications, and it can also be used by researchers who want to learn more about pH-responsive MOFs. ZIF-8’s size and functionalization, on the other hand, should be evaluated from a biological point of view, as its morphologies influence cancer cell ingestion rates. However, the intricacy of the human body‘s environment restricts the capacity of single stimulus-responsive Zn-NMOF drug carriers to accurately distribute pharmaceuticals in the body [94]. In order to counter the aforementioned disadvantage, different Zn-based carrier systems were adopted with various stimuli-responsive as a superior option to boost delivery capacities and chemotherapeutic efficiency. In this study, they concentrate on the subject of pH-responsive agents based on Zn-based MOFs that has emerged in the past 20 years. UCNPs@ZIF-8/FA had a 5-FU extraction efficiency of up to 71% to 82% at approximate pH 5.5 after 12 and 24 h equally in an in vitro drug release experiment over pH 7.4; however, only 35% was reported when ZIF-8 quickly dissolved in the acid medium [95].

### 3.2. pH-and Temperature Responsive Zn-MOFs Nanocarrier

Lin and his team developed a pH and temperature-dependent drug release framework with one-dimensional channels and from constituents such as zinc ions and bifunctional ligands. The developed framework, Zn-TBDA (TBDA ¼ 4′-(1H-tetrazol-5-yl)-[1,1′-biphenyl]-3,5-dicarboxylic acid), showed insignificant cytotoxic activity and depicted an appropriate methotrexate holding potential and controlled release. The dual temperature and pH responsive release of encapsulated drug of this complex metal organic framework established its ability as an efficient drug/medicinal carrier [96]. Another zinc based MOF was prepared with tricarboxylic ligand for the storage and delivery of 5-fluorouracil and the resultant structure represented appropriate window size, better BET surface area, and improved density of open free N sites. The developed framework showed pH responsive behavior by releasing 68.6% of the stored 5-fluorouracil at a pH of 6.5 that was increased to 86.5% at pH of 5.0 after 24 h. Additionally, the toxicity of synthesized framework was also measured for HepG2 and HFL1 by performing MTT assay, and it showed satisfactory results [97].

Doxorubicin (DOX) has been extensively explored as a standard anticancer treatment [98,99]. There was a one-pot procedure that was used to make homogeneous nanoscale DOX@ZIF-8. DOX@ZIF-8 remained stable after 7 days at 60 °C and pH 7.4 diluted PBS solutions, while the ZIF-8 carrier discharged around 90% of DOX at pH 5.5 because its framework became denatured in high pH solutions. DOX@ZIF-8 was more effective against human breast cancer cells versus free DOX in the MTT testing. In viable cells, the MTT test demonstrated no cytotoxicity, revealing that the NPs might be employed as biodegradable DDS in the next. 

A one-pot encapsulation procedure was used to create the core-shell nanostructure utilizing BSA/ DOX and ZIF-8 [100]. The fundamental nanostructure protected the activity of BSA, which increased the efficacy of DOX and reduced adverse medication effects. The addition of positive charges to BAS/surfaces DOX’s will also boost its cellular absorption. Similarly, drug release assays revealed that only 84% of the pharmaceuticals were expelled after 1 day at pH 7.4; on the other hand, they discharged nearly all of the medicines at about the same period at pH 5.5. Finally, the pH-responsive characteristic of the foundational ZIF-8 substance permitted it to regulate the untimely emission of DOX in defined circumstances. BAS/DOX@ZIF-8 outperformed pure DOX, BAS/DOX, and ZIF-8 in regards to biomedical applications and lethality, which can be attributable to their complex formation. With pH-responsive properties and antitumor medical therapies, the novel fundamental DDS will become an exciting technology. One method also has been used to create an ionic strength ZnO-DOX/ZIF-8 DDS to control early drug delivery and improve medicine therapeutic effictivness in tumor tissue. 

At pH 7.4, about 20% of DOX was liberated through DDS, although in the pH 5.5 buffer, it was predicted to be around 84 %. Due to the fact that the DOX with radical oxidation products generated synergic activity, ZnO-DOX/ZIF-8 exhibited higher cellular uptake and lethality over DOX as well as ZnO@ZIF-8. These results pointed to the ZnO-DOX@ZIF-8 transporter as an effective pH-responsive drug carrier strategy for treatment for cancer. New functionalized DOX@ZIF-8 NPs enabling endo/lysosomal egress with pH-responsive liberation in the cellular environment were discovered [101].

The aptamer (AS1411) transplanted on the ZIF-8 may boosts malignant cells absorption while minimizing DOX negative consequences. The customized aptamer also exhibited great pH-responsive drug dissolution capabilities, with only 5% of DOX discharged from NPs within 24 h under PBS (pH 7.4) and 50% pharmaceutical unloaded in highly acidic (pH 5.0). Pursuant to in vitro drug experiments, NPs liberated 22.5% and 26.72% of DOX at pH 7.4 around 12 and 24 h, accordingly, while 47.92% and 55.1% were liberated at pH 5.5 at the same time. This is also attributed to the pH-responsive chitin object’s ability to alter dose forms over time, as well as the effect of extreme pH on IRMOF-3 breakdown. Due to Fe_3_O_4_’s paramagnetic feature, NPs can be employed for magnetic resonance imaging (MRI), indicating that they can be used as a permanent dipole therapeutic agent. Investigators conceived and manufactured a co-delivery, pH-responsive, and highly directed multipurpose medication transporter incorporating DOX/verapamil hydrochloride (VER)@ZIF-8 methoxy polyethylene glycol)-folate (PEG-FA) to improve DOX efficacy.

The DOX carrying capacity of the carrier is 40.9%. Furthermore, while VER works as a P-glycoprotein regulator to eliminate FA’s multi-drug sensitivity and specific function in vivo and in vitro, its clinical efficacy is significantly greater than among free DOX as well as other DOX@ZIF-8. Meanwhile, in pH-responsive discharge experiments, DOX and VER exhibited substantial consistent characteristics. A new Fe_3_O_4_@PAA/gold nanoclusters (Ounces)/ZIF-8 system was manufactured by using a simple and gentle technique. It can also help with magnetic resonance imaging (MRI) and X-ray computed tomography (CT), fluorescent quadra scanning, and also have an ultra-high DOX load-bearing capacity. DOX was found to have loading efficiencies of approximately 1.54 *g*/*g*. Moreover, over 68% of the injected medication has been delivered from the medium during 26.5 h with pH 5.3 due to nucleophilic attack of the carboxylic acid groups by DOX and dissolution of ZIF-8 compared to only 15% at pH 7.4 at the same time, showing the double pH-responsive functionality of such a transporter.

Based on in vitro drug release studies, AP-ZIF-90@DOX emitted only 1.7% DOX at pH 7.4 but 19.8% once 0.5 mM ATP was introduced. Over 21.7% DOX also generated within 2 h with pH 5.0 with 0.5 mM ATP, prompting ZIF-90 to disintegrate under acidity and ATP circumstances. AP-ZIF-90@DOX demonstrated good bioactivity with the animal model after 1 month of in vitro and in vivo mortality studies [102].

In vitro and in vivo mortality experiments demonstrated that Fe_3_O_4_@PAA/AuNCs/ZIF-8 NPs were both nontoxic and anticancer. The multifunctional world features a lot of possibilities for investigations, chemotherapy, and double pH-responsive drug delivery due to the evident portion of the multi-imaging of Fe_3_O_4_@PAA and AuNCs. ZIF-90, which likewise possesses high cell compatibility and pH-responsive drug-release activity, is compensated by zinc ions and imidazole-2-carboxaldehyde (2-ICA). The surface of ZIF-90 was changed by a displacement-based interaction between the aldehyde group of 2-ICA and the amino group of DOX, and 5-FU was subsequently loaded by ZIF-90 to produce a new nanotechnology medicinal carrier [103]. According to medication loaded data, the loading levels of 5-FU and DOX were 36.35 percent and 13.5 percent, respectively. The substance activity of ZIF-90 was also ionic strength sensitive, with 44 percent 5-FU and 20 percent DOX freed in PBS at pH 7.4 after 25 h, while 95 percent and 91 percent of drug release occurred with pH 5.5 after 15 h and 25 h, respectively. The researchers also created a pH-responsive targeted Nano carrier by combining the Y1 receptor ligands [Asn, 6 Pro34]-NPY (AP) on the interface of ZIF-90 by inserting DOX into ZIF-90 holes. The payload successfully identified and treated triple-negative breast cancer cells.

One-pot wrapping boosted medication availability by roughly 12.6 percent in the loading DOX experiment. Due to the fact that the physical interaction between adenosine triphosphate (ATP) and Zn^2+^ is substantially stronger than physical bonding between imidazole and Zn^2+^, AP-ZIF-90-DOX exhibits dual ATP and pH-responsive drug-releasing capabilities.

According to in vitro drug release tests, AP-ZIF-90@DOX released only 1.7 percent DOX at pH 7.4 but released 19.8 percent once 0.5 mM ATP was added. More than 21.7 percent DOX was also produced within 2 h at pH 5.0 with 0.5 mM ATP, causing ZIF-90 to dissolve under acidic and ATP conditions. After 1 month of in vitro and in vivo mortality investigations, AP-ZIF-90@DOX revealed amazing bioactivity using an animal. 

UC@mSiO_2_-RB@ZIF-90-O_2_-DOX-PEGFA (UC = NaYF4:Yb/Er@NaYbF4: Nd@NaGdF4, RB = Rose Bengal) has also been created as a pH-responsive multifunctional O_2_-loaded UC@mSiO_2_-RB@ZIF-90-O_2_-DOX-PEGFA (UC = NaYF4: Yb/Er@Na). When exposed to laser light, the core-shell structured UC triggered the photosensitizer (RB) in mSiO_2_ to generate ROS for PDT. ZIF-90 acts as an oxygen buffer, producing oxygen free radicals that aid in the treatment of malignant hypoxia, improve PDT effectiveness, and demonstrate pH-responsive DOX releasing activity in the tumour. Furthermore, NH_2_ PEG-modified folic acid (PEGFA) was directly coupled mostly on exterior drug ZIF-90 to improve selective administration and optimise clinical efficiency, and the amount of DOX generated ranged from 79.8 percent to 7.1 percent when the pH was adjusted from 5.5 to 7.3 after 17 h within pH-responsive and DOX release investigations. Based on the effective load of O_2_-loaded PDT and DOX, UC@mSiO_2_-RB@ZIF-O_2_-DOX-PEGFA showed exceptional anticancer treatment efficacy in in vitro and in vivo studies [104].

### 3.3. Design of pH-Responsive/Dependent Drug Carriers

It is a remarkable achievement to design pH-dependent medicinal carriers in Zn-based MOFs for medical applications. The most commonly used zinc-based MOF for the pH-dependent release of medical drugs is ZIF-8. It has a sodalite (SOD) structure due to its porosity (11.6 Å) and sensitivity to acid. Correspondingly, the drug is depicted to have better medicine carrying capacity and pH-dependent release. Moreover, polyacrylicacid@ZIF-8 (commonly known as PAA@ZIF-8) possessed improved DOX loading capacity of 1.9 *g*/*g*. However, this is attributed to electrostatic reactions occurring between positively and negatively charged DOX and PAA and coordination interaction among Zn^2+^–DOX. Therefore, the medication loaded with MOF was found to be released faster into a buffer solution at pH 5.5 rather than at neutral pH by confocal laser scanning analyses [105]. Moreover, scientists developed a pH and temperature-dependent drug release framework with one-dimensional channels from the constituents of zinc ions and bifunctional ligands [106]. For example, framework Zn-TBDA (TBDA ¼ 4′-(1H-tetrazole-5-yl)-[1,1′-biphenyl]-3,5-dicarboxylic acid) showed insignificant cytotoxic activity and depicted an appropriate methotrexate holding potential and controlled release. The dual temperature and pH-responsive release of encapsulated drugs of this complex established its ability as an efficient drug carrier.

Another zinc-based MOF with appropriate size, better BET surface area, and improved density for open free N sites was prepared with tricarboxylic ligand for the storage and delivery of 5-fluorouracil. This framework showed pH-responsive behavior by releasing 68.6% of the stored 5-fluorouracil at 6.5 pH, further increasing to 86.5% at 5.0 pH after 24 h. Additionally, the toxicity of this synthesized framework for HepG2 and HFL1 was also measured by MTT assay, which showed satisfactory results.

## 4. Medical Applications

### Zn-Based MOF as Drug Carriers

Many medicinally active ingredients are confined to limited numbers in pharmaceutical applications due to poor stability in living systems, low solubility, and inability to cross natural barriers [2]. Development in the pharmaceutical industry started in the 1970s by introducing drug Nano carriers to improve techniques to prevent the biodegradation of active ingredients, thus protecting the living system from their toxic effects. These techniques improved the effectiveness of the drug by increasing intracellular penetration. Furthermore, Nano techniques have also paved the way for targeting specific tissues, cells, and even cellular organelles [52]. Zinc-based metal-organic frameworks are considered one of the best candidates for developing Nano encapsulates due to their versatility, lower toxicity, and being easily biodegradable.

Different MOFs based on Zn with their encapsulated drug, organic linker, loading degree, and release rate are presented in Table 1. Likewise, a zinc-based MOF was established with bidentate carbene ligand as rigid support that makes it possible for Zn-MOF to assist in encapsulating and then discharge the drug (cisplatin). Cisplatin, an effective anticancer medicine, has been verified for its anticancer potential against ovarian cancer cells (A2780) through in vitro trials.

Extremely water-stable (up to 3 weeks), microporous MOF, [Zn8(O)2(CDDB)6(DMF)4(H2O)] where CDDB = 4,4′-(9-H carbazole-3,6-diyl) dibenzoic, was synthesized by a solvothermal process based on an open N-H site, which demonstrated excellent loading potential (around 53.3 wt.%) and adequate release potential (64.9 and 81.9%) for 5-fluorouracil, and loading capacity is around 53.3 wt%.

An odd 4,8-connected 3D net with (46)2(412•612•84) topology was shown by the coordination polymer. The medication 5-fluorouracil (5-FU) was connected to the desolated one at around 22.5% wt. per gram of the dehydrated one. With 92% of the prescription released after 120 h, 5-Fu is discharged in a closely regulated and regular manner. This analysis provides a new method for MOF to be used as possible drug delivery. Compared to zeolites and activated carbons, MOF showed both hydrophilic/hydrophobic entities, and their versatile nature allows porosity to be tailored to physicochemical characteristics. In comparison, structural instability in water has been observed in specific Zn-based metal-organic frameworks. However, to overcome this issue, a biofriendly and water-stable anionic Zn-MOF has been prepared with a dicarboxylic acid as the organic linker.

This medication had a 25% drug loading weight and 95.6% removal efficiency after 100 h. The non-toxicity of the synthesized drug has been testified by conducting in vitro MTT assay and in vivo toxicity tests. Drug loading in water (53.3% by weight and release rate of 81.9%) and 64.9% in PBS was achieved for 5-fluorouracil with negligible cellular toxicity.

## 5. Applications of Zn-Based MOFs for Gases Adsorption, Imaging and Sensors

Zn-based MOFs have added beneficial aspects in drug delivery as efficient medicinal carriers; they also have other applications. Here, we will summarize the applications of MOFs in various disciplines [107]. 

### 5.1. Zn-Based MOFs for Active Gases Adsorption

Zn-based MOFs are capable of capturing, separating, storage, and effective delivery of bio medically essential gases. Nitric oxide is an exceptional biomolecule with antibacterial, anti-thrombosis properties and an excellent vasodilating agent [108]. In progressive studies, the capability of Zn^2+^ exchanged zeolite: polytetrafluoroethylene (50:50) for nitric oxide loading, release, and antibacterial properties was examined. They found better antibacterial activity for nitric oxide-releasing zeolites, whereas nitric oxide-free Zn^2+^-exchanged zeolite provided excellent potential for fabricating extremely operative dual functionality-bacterial materials, while the application of such nitric oxide donors might include experienced confinement due to the production of certain pro-inflammatory by-products [109].

However, metal-organic frameworks acquire good potential because of their improved storage and control over reaction ability between gas molecules and components of MOF by making appropriate alterations in components. Thus, a Zn-based MOF has been developed by incorporating tetracyanoquindimethane as a linker to create a porous structure for selective absorption of nitric oxide and other gas CO_2_, N_2_, C_2_H_2_, and molecules [110]. CPO-27 metal-organic frameworks exist in the form of iso-structures formed by the combination of 2-5-dihydroxyterephthalic acid with different metallic parts, including Mg, Zn, and Ni [111]. These clusters have pore sizes ranging 11–12 Å and could be employed to store and release nitric oxide. Furthermore, Ni^2+^ doped CPO-27 (Zn) has been used to strengthen CPO-27 (Zn) in order to enhance future nitrous oxide release in nitric oxide retention. This composite possessed good thermal stability and the efficient ability to synchronize the solvent molecules, freely subjected to heat to create open sites to capture the nitric oxide [112].

Hydrogen sulfide (H_2_S) is also an extremely toxic gas; it could be biologically significant if delivered in a required amount at the desired rate. It has been discovered that metal-organic frameworks with the CPO-27 structure can be used to improve H_2_S adsorption and preservation. Correspondingly, H_2_S binding and releasing capacity of Ni-CPO and Zn-CPO has been testified, demonstrating better nickel delivery (i.e., 1.8 mmol·g^−1^ from nickel and 0.5 mmol·g^−1^ from Zn in 30 min) as compared to Zn-based material. However, the ability to deliver less H_2_S from Zn-based material could be linked with difficulties, such as the incomplete activation of the sample, as earlier reported, or the irreversible and substantial binding potential of zinc to sulfur [113].

### 5.2. Zn-MOF as Contrast Agents in MRI

MOFs have recently been used to replace contrast chemicals in biomedical imaging modalities such as magnetic resonance imaging (MRI). Magnetic resonance imaging (MRI) is widely acknowledged as one of the most prospective imaging methods that do not require ionizing radiation or a radioactive nucleus. Magnetic nanoparticles, on the other hand, can change the MRI signal by speeding up the relaxation phase of protons in nearby H_2_O molecules under an outside magnetic field [114]. Research studies on the fabrication of MOFs as MRI contrast agents have been ongoing since the mid 2000s [115]. The recent developments on Gd III-based nanoscale with good paramagnetic properties have enabled their application as MRI contrast agents.

A versatile framework for magnetized derived medication delivery, MRI agents, and biomedical imaging has also been established using an iron-based MOF nanoscale; two major applications of MOFs include the following: drug carriers for controlled release and the potent contrast agent in MRI image augmentation. ZIF-8 is one of the most extensively studied Nano carriers, emphasizing its application in cancer imaging and treatment. Progressively, glutathione and pH-dependent ZIF-8 were employed to serve as platforms for assembling small Fe_3_O_4_ nanoparticles as T1 contrast agents in the assembly of Fe_3_O_4_–ZIF-8 acting as a T2 contrasting agent [116]. In vivo investigations used MRI contrast agents with Fe_3_O_4_-based T2–T1 switching for cancer diagnostics. To replace 2-methylimidazolate in ZIF-8 [117], a pH-sensitive 19F MRI Nano probe was created. With a low background and excellent penetration depth, this probe was well suited for stimuli-responsive detection. However, due to their toxicity, these materials are not suited for biotechnology. Nanoparticles have indeed been created for effective delivery systems by combining biosynthetic pathways and scanning features with MOFs. A strong MR-active saturation magnetization micro-MOF linked with folic acid as a fluorescent marker has been efficiently produced that delivers aqueous anticancer medication paclitaxel (Figure 4). These Fe_3_O_4_@IRMOF-3/FA NPs are a multipurpose spongy framework that could be used for medication administration, bio imaging, and MRI contrast [118].

However, due to their toxicity, these materials are not suited for biotechnology. Nanoparticles theranostics have indeed been created for effective delivery systems by combining biosynthetic pathways and scanning features with MOFs. A strong MR-active saturation magnetization micro-MOF linked with folic acid as a fluorescent marker has been efficiently produced that can deliver the aqueous anticancer medication paclitaxel. These Fe_3_O_4_@IRMOF-3/FA NPs are a spongy multipurpose framework that could be used for medication administration, bio imaging, and MRI contrast. Regarding cancerous cells, concatenated folic acid demonstrated a greater T2-weighted MRI distinction. ZIF-8, a more investigated nanoparticles that has been employed in clinical therapy and diagnosis, is a much more investigated nanoparticle. Small Fe_3_O_4_ NPs (T1 distinction agent) have indeed been produced together into T2 contrast material Fe_3_O_4_–ZIF-8 architecture that used a pH and glutathione-responsive ZIF-8 as both structures. For in vivo diagnostic procedures, an Fe_3_O_4_-based precise T2–T1 switching image reconstruction approach is particularly successful. Since 1H MRI has a low exposure, the number of protons in human tissue causes strong background signals.

### 5.3. Nanoscale Metal-Organic Framework for Ultrasonographic Scanning

Nanoscale MOFs have been investigated for their possible use in a variety of applications such as X-ray computed tomography imaging (CT imaging) as contrast agents with the addition of high atomic number elements as their building blocks [119]. Meanwhile, Cu^2+^ and Zn^2+^ were employed as metal parts along with 2,3,4,5,6-tetraiodo-1,4-benzenedicarboxylate (I4-BDC) as bridging ligands for the fabrication of iodinated nanoscale metal-organic frameworks [Cu(I4-BDC) (H_2_O)2] ·2H_2_O and [Zn(I4-BDC)- (EtOH)2] ·2EtOH, respectively. These fabricated MOFs contained a high level of 63% of iodine weight. In contrast, phantom studies verified that X-ray attenuation coefficients of these MOFs are analogous to the conventionally used molecular contrast agent such as iodixanol. Therefore, it can be concluded that these frameworks provided a unique platform for devising efficient contrast agents that could be used in CTI by incorporating iodinated compounds as bridging ligands [120].

### 5.4. Chemosensors Composed of Zn-Based MOFs

There has been extensive research being conducted regarding applications of biosensors for detection and quantification of certain compounds. Biosensor is a term used to define a device with sensing elements of biological origin, which is linked to or assimilated within a transducer [121,122].

Metal organic frameworks that are constituted by assembling metal and organic linkers have been studied broadly as chemical sensors. The most imperative characteristic of these MOFs is their inherent luminescence that make possible their application as sensors. In recent times, nanomaterials have been broadly investigated as bio or chemo sensors with improved selectivity, sensitivity, and response time because of their certain chemical properties attributed to particular host–guest interaction [123]. Zinc based MOFs might be transduced into a device because of their luminescent characteristics and, hence, act as chemical sensors; therefore, they are favorable ligand for the synthesis of Zn-MOFs having nucleophilic attributes in effective luminescent sensors and could be electron-rich’ p-conjugated fluorescent compounds [124]. In addition, the organic linkers in such MOFs are possible to fabricate on the basis of their potentials in oxidative quenching. Luminescent lanthanide cations, which emit in NIR and visible regions, were introduced into the bio-MOF-1 pores for biomedical applications [125] (Figure 5). This inflexible anionic porous framework (1700 m^2^/g) can be utilized as a photostable O_2_ sensor. In another study, temperature and luminescent sensors were successfully deliberated by employing [2,4,6-tris(3,5-dicarboxyl phenylamino)-1,3,5-triazine] as an organic linker in MOFs with Zn as metal. An in situ doping approach can be applied for achieving MOFs with magnetic and fluorescence attributes. In this approach, Rexit created a heptanuclear zinc cluster-based MOF with luminescence using a polycarboxylate ligand containing trifluoromethyl groups. Temperature and luminescence sensors were effectively developed in another work by using [2,4,6-tris(3,5-dicarboxylphenylamino)-1,3,5-triazine] as an organic linker in MOFs with Zn as the metal [126].

The glycopeptides of human blood (1 L) were enriched after incubation with the MG@Zn-MOFs, followed by elution, deglycosylation, and independent analysis using nano-LC-MS/MS. Surprisingly, 517 distinct N glycopeptides and 151 distinct glycoproteins may be effectively identified. Magnetic graphene hydrophilic porous bio composite (MG@Zn-MOFs) functionalized with metal-organic frameworks (MOFs) for glycopeptide recognition. MG@Zn-MOFs demonstrated great sensitivity and selectivity, as well as high recyclability, in glycopeptides analysis due to their strong magnetic responsiveness, large specific surface area, superior biocompatibility and unique size-exclusion effect. In practice, 517 N-glycopeptides among 151 distinct glycoproteins were definitely identified from human serum (1 L) treated with the MG@Zn-MOFs, which is the best result among published results so far. The MG@Zn-MOFs biocomposite outperforms the individual magnetic graphene and Zn-MOFs counterparts in glycopeptide selectivity. Furthermore, the developed biocomposite exhibits remarkable results for glycopeptide enrichment in human blood (1 L), demonstrating its potential use in practical applications. The MG@Zn-MOFs biocomposite combines the benefits of magnetic graphene and Zn-MOFs, including a large number of affinity sites, a unique aperture-screen effect, great magnetic responsiveness, and a high specific surface area. Graphene-based materials have also been employed in the enrichment of glycopeptides. Wang et al. developed an MOF-functionalized magnetic graphene nanoporous composite (C-magG@ZIF-8) that demonstrated high selectivity, sensitivity, and repeatability. MOFs are synthesized using a crystal assemble growth technique by grafting Zn-MOF onto magnetic graphene sheets, resulting in homogeneous coating. The major properties include a large surface area, high stability, and an improved size-exclusion effect. Using this affinity material, 517 N-glycosylation sites in human blood are found, which is the most so far in glycoproteome study [127,128]. 

Saleem, Shafaq, and colleagues created a hydrophilic terpolymer MOF composite with a large surface area and porosity to enrich mono-glycosylated and multi-glycosylated peptides, allowing for a bottom-up approach. Terpolymer@ZIF-8 is created by using free radical polymerization and layer-by-layer ZIF-8 manufacture. Aminophenylboronic acid was then used to modify the surface (AMBA). After exposing enriched and PNGase-treated deglycosylated peptides to LC-MS, 318 N-linked glycopeptides were discovered from 1 L human serum digest. As a result, terpolymer@ZIF-8@BA has the capacity to produce mono-glycosylated and multi-glycosylated peptides from complicated biological samples [129].

Sawayama, Taku, and colleagues discovered that metal-organic frameworks (MOFs) can distinguish topological designs of macromolecules, such as linear and cyclic structures, by selective polymer insertion into nanopores. They described a novel polymer separation technology that use metal-organic frameworks (MOFs) to distinguish between linear and cyclic polyethylene glycols (PEGs) by selective polymer insertion into the MOF nanopores. Analytical and preparative chromatographic separations of these topologically different pairings ere made possible by the preparation of an MOF-packed column. Furthermore, utilizing MOF as an adsorbent, gram-scale PEGs with solely cyclic structures were effectively produced from a crude reaction mixture. Oe and Noriyoshi et al. revealed that polymer insertion into sub-nanoporous MOFs happens spontaneously from the solution phase [130]. When the system achieves sufficient enthalpy to overcome potential entropic losses owing to uncoiling of the solvated polymer chains, admission of polymer chains that assume coiled and self-entangled conformations with exceptionally higher hydrodynamic diameters relative to the pore diameter is achievable. PEGs were used to explore the thermodynamics and kinetics of polymer insertion, which revealed an enthalpy-driven mechanism coupled with a dynamic insertion/rejection process at the solution/MOF interface. Isothermal adsorption study indicated a unique solvent dependency of PEG insertion, which is explained by PEG-MOF affinity, MOF-solvent affinity, and the solvation effect [131]. 

We proved the significant potential of MOFs for the separation of polymers that vary only in their terminal groups, in addition to their utility for the separation of gases and small molecules. In this method, polymer chains are inserted into MOFs through their extremities, allowing for the separation of functionalized PEG with diverse terminal groups of varying natures and sizes. It was interesting to see how MOFs with stiff pores could distinguish the terminal groups of PEG based on steric hindrance. Furthermore, we could take use of the dynamic nature of flexible MOFs to precisely recognize terminal groups of comparable size but slightly different polarity. These two separation modes, which are critical for the separation of tiny molecules by MOFs, retain their capacity to distinguish terminal groups even when they are connected to lengthy polymer chains in this case. Thus, the utilization of MOFs may be at the heart of a highly wanted technology for the robust and systematic separation of polymers, particularly PEG, for which their pervasive usage in biology necessitates the use of very pure materials. Due to the diversity of MOFs, our technique may be expanded to a broad range of polymer separations, offering up interesting possibilities for highly adaptable polymer purification systems [132].

Furthermore, the organic linkers in MOFs can be customized via the oxidative quenching method. For sensing nitro aromatics, particularly trinitrophenol (TNP), zinc-based luminous MOFs were synthesized using a flexible electron-rich N-involved linker [133]. Photo-excitation power is transported from the linker to the cations TNP when anion MOFs engage with TNP, resulting in luminescence quenching. The in situ doping method may account for both magnetic and fluorescence characteristics.

This was demonstrated by synthesizing a heptanuclearzinc cluster-based MOF with luminescence properties based on a polycarboxylate ligand containing trifluoromethyl groups. Poorer fluorescence characteristics are observed in the Co^2+^-doped Zn^2+^-MO. In another place, TDPAT [2,4,6-tris(3,5-dicarboxyl phenyl amino)-1,3,5-triazine] has been used as a bridge for Zn-based MOFs to build an effective fluorescence or temperature monitoring agent (Figure 6a,b) [126]. 

### 5.5. MOFs for Antigens and Adjuvants 

Due to the fact that antigens have limited immunogenicity and little binding ability to the target, using them as vaccines on their own typically results in modest immune responses [134]. Due to their great loading capacity, nMOFs may assist antigens in directly targeting the immune response. These compounds are varied in composition and structure and are readily adjustable, making it easier to make nMOFs with desired forms, sizes, and chemical characteristics for any sort of targeting, including lymph node [135,136,137]. ZIF-8’s imidazole was protonated inside acidic lysosomes, changing its interaction with zinc ions and enabling OVA to escape from lysosomes into the cytoplasm and be cross-presented by dendritic cells (DCs), as well as a larger percentage of CD8+ T cells secreting IFN-+ and IgG2a antibody. The OVA and CpG combination produced less secretion than the nanoparticles. In mice carrying EG7-OVA cells (EL-4 thymoma tumour cells transfected with the OVA gene), CpG/ZANPs dramatically decreased tumour development and extended life, demonstrating increased antigen-specific CTL response.

Based on these findings, nMOFs have the following advantages: (i) extremely high antigen loading capacity; (ii) nMOFs can subsequently enhance antigen immunogenicity and uptake by APCs; (iii) nMOFs degraded in acidic endo/lysosomes enable the release of encapsulated antigens into the cytoplasm, resulting in enhanced antigen cross-presentation ability; and (iv) compatibility.

## 6. Problems Associated with Current Zn-MOFs

The utility of Zn-MOFs in medical treatments is hampered by various factors. However, these factors are not unsolvable but could be a barrier for actively progressing scientific inventions. However, some of the essential elements contributing primarily to the limitations of Zn-MOFs application are summarized with their possible remedies in the following sections.

### 6.1. Stability

Various factors such as host–guest interaction, toxicity, porosity, hydrophilic, and hydrophobic ratios are responsible for implementing MOFs. Nevertheless, performance in the applied system intensely depends on their biodegradable characteristics. However, the framework is based on the bonding forces between metallic and organic ligand parts. An uninterrupted ligand interchange phenomenon is expected from the complexing group to the water molecule from the metal and vice versa. Therefore, while working with MOFs in biological systems with diverse composition ranges and pH, it is imperative to look for the stability of these frameworks, which impact their activity, while stability measurements could be carried out by exposing MOFs to different conditions that are very much similar to the biological ones such as phosphate buffer solution, simulated body fluids, and water, followed by determination of any defect in the MOFs structure without any significant degradation [138]. At the same time, the instability of MOFs structure in the presence of water may reduce their activity by affecting their pores volume and internal surface.

Various research studies reported that MOF-5, also known as zinc terephthalate, is extremely unstable in water. In further studies, MOFs based on Zn metal is reticular MOF-1, known as IRMOF-1, having a porous cubical structure that showed appropriate stability up to 450 °C in the presence of nitrogen [139]. It suffers irreparable structural degradation in the presence of atmospheric moisture [140]. Nitrogen bases containing MOFs with organic linkers and zinc as metal parts have improved stability compared to those constituted from irreparable structural degradation [141].

The problem can be addressed by applying different strategies for improving the stability of the MOFs such as functionalization and selection of appropriate composition and structure. Thus, a significant factor to be well thought-out in application of MOFs is modification in their surface and control over distribution of particle size. Not only can the stability of MOFs be enhanced by such surface alterations but also drug carrying procedure through diverse physiological barriers [24].

In the recent times, researchers have paid attention to the modification of MOFs for improving their stability and, in this regard, the development of nano MOFs or alterations in MOFs with metals in the form of nanoparticles is an important milestone. The activity of such MOFs is dependent upon properties of nanoparticles and the particular type of linkers. An example of a nanoparticle-based MOF is the ZIF-8 with particles of controlled size. Poly (ethylene glycol) (PEG) surface modification has been employed to control the particle size of ZIF-8 by using PEG as a capping agent. These MOFs revealed pH dependent drug release efficacy against cancer cells.

Modification in the structure of MOFs could be performed according to the required application type by dispersing MOFs particles inside their matrix or could be accomplished by forming composites in core-shell [142]. Core-shell composite, i.e., IRMOF-2@MOF-5 can carry the load and could be the potential host for guest compounds [143]. Another modification in MOFs is the use of bimetallic MOFs, which exhibit improved framework stability as compared to their single parent MOFs [49]. An example of such MOFs is the development of ZnZr-based bimetallic MOFs by using MOF-on-MOF techniques, and it was testified as aptasensor scaffolds for sensing cancer marker protein tyrosine kinase-7 (PTK7). These complexes were formulated by altering various orders of metal precursors as well as ligands. The improved activity in bimetallic MOFs is achieved as Zr-MOFs assist aptamer immobilization, while the Zn-based MOF caused the stabilization of the G-quadruplex formed by aptamer strands and PTK7 [144,145].

### 6.2. Toxicity Studies

The wide range of available components for the design of MOFs may allow finding the appropriate one with toxicological acceptability. MOFs that are based on linkers from endogenous sources and metals are limited as compared to general MOFs [99]. In toxicological assessments, complete composition of MOFs must be subjected to different assays to assess the incidence of any toxic molecule (i.e., ligand or metal) in their structure. In this regard, the application of porous MOFs for biomedical application will require preliminary toxicological assessment trails for each of the components [24]. Zn, among others toxic and abundantly used metal, is comparatively safe in biological functions and, hence, considered suitable for theranostic applications. The studies related to the toxicity and stability of various Zn-based metal organic frameworks conducted with different human cell lines and various in vitro assays are presented in Table 2, which include estimating the cytotoxicity of one of the Zn-based MOF, i.e., nanoZIF-8 for three human cell lines such as H292, HT-29, and HL-60. It was concluded that nanoZIF-8 poses no cytotoxicity to any of the above-mentioned cells at 109 mm (IC50) [146,147].

## 7. Conclusions

Current data about development and applications of MOFs are quickly proliferating in recent years, but there is still a noteworthy gap in the complete understanding of their structure, properties, and stability under different conditions. There is room to investigate in-depth mechanisms responsible for the destruction of crystalline structure of these MOFs. In areas of biomedical applications, most commonly used drugs are mainly composed of anticancer and antiviral molecules. Such applications can be further broadened to carry drugs against other disorders as well. A comprehensive in vitro and in vivo toxicity assessment is mandatory for commercialization of such products. Another advancement in the development of MOFs is the creation of bio-MOFs from constituents such as biocompatible metals and activated pharmaceutical ingredients (APIs) followed by their degradation within physiological surroundings that allow for the gradual release of loaded drugs like that of encapsulated APIs within the pores of traditional MOFs. Both methods can be combined as well for codelivery of the drug to the targeted site either as a component of framework or encapsulated within pores. In the future, MOFs can also be tested for the delivery of larger, complex biomolecules such as hormones [150,151].

In vivo studies to investigate the stability, degradation, and processes of MOFs are currently missing. Although there is a review article on the uses of Zn-based MOF in drug delivery and cell imaging, most of the Zinc-based metal-organic frameworks as nontoxic and biodegradable platforms for biomedical applications are yet unknown. However, to our knowledge, no recent review paper focusing on the applications and synthesis of Zn-based MOF has been published. As a result, a review of the many uses of various Zn-based MOF with various linkers is required to provide a critical assessment of the data available from previous studies as well as to identify gaps for future study. This study aims to provide the most up-to-date information on the uses of various Zn-based MOFs. Furthermore, the review discusses toxicity, stability, and effects of pH on the application of MOFs, as well as the efficacy of different Zn-based MOFs for drug delivery and cell imaging, as well as the challenges and evidence gaps that arise when using different linkers to make the application of MOF more effective, as well as suggesting knowledge gaps to researchers for future studies [152]. Various studies using multimodal imaging methods have been carried out to monitor the degrading routes of MOFs, but further research is needed to fully understand the process. This cleared the door for further research into MOF routes in the future. In this context, the long-term absorption–distribution–metabolism–excretion pathway must be carefully monitored to assess the accumulation of MOFs in various tissues due to the diverse modes of administration. Scale-up development is still a barrier to mass manufacturing for this innovative family of porous crystalline materials. As a result, the world of MOFs has to be further researched in order to attain cost efficiency and improved performance, as well as high purity and yield of nanostructures.

## Figures and Tables

**Figure 1 molecules-27-00100-f001:**
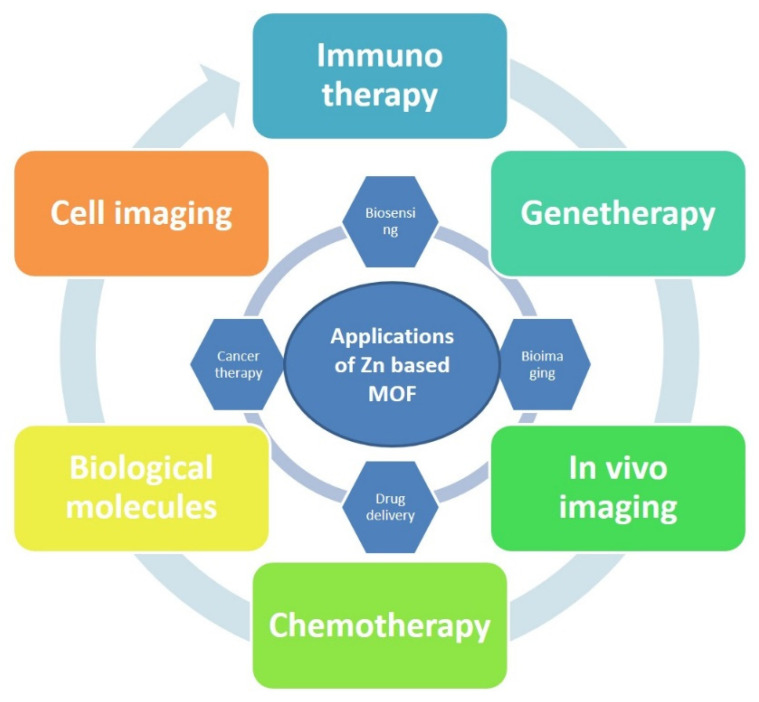
Summary of Zn-based MOFs applications.

**Figure 2 molecules-27-00100-f002:**
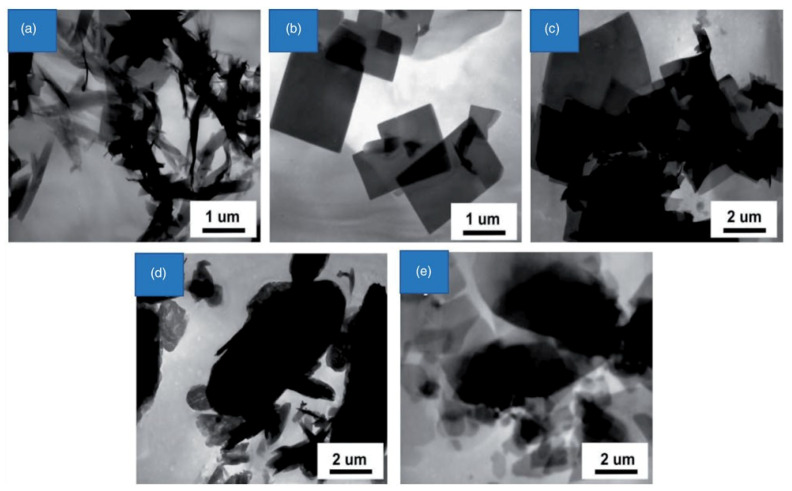
The transmission electron microscopic (TEM) result of [Zn(BDC) (H_2_O)]n following (**a**) 10, (**b**) 20, (**c**) 30, (**d**) 60, and (**e**) 90 min sonication [61]. Reprinted with permission from Ref. [61]. Copyright 2008 Elsevier.

**Figure 3 molecules-27-00100-f003:**
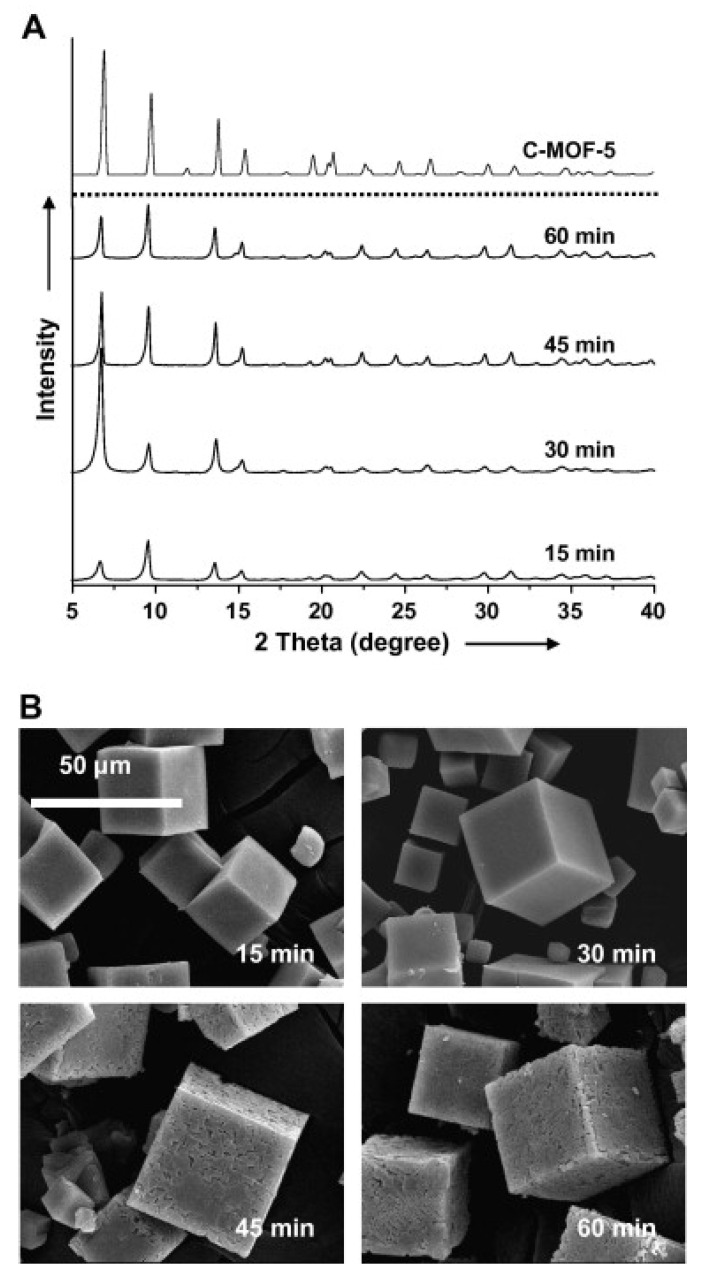
(**A**) XRD and (**B**) SEM images of MMOF-5 samples: microwave power was supplied at 600 W [72]. Reprinted with permission from Ref. [72]. Copyright 2008 Elsevier.

**Figure 4 molecules-27-00100-f004:**
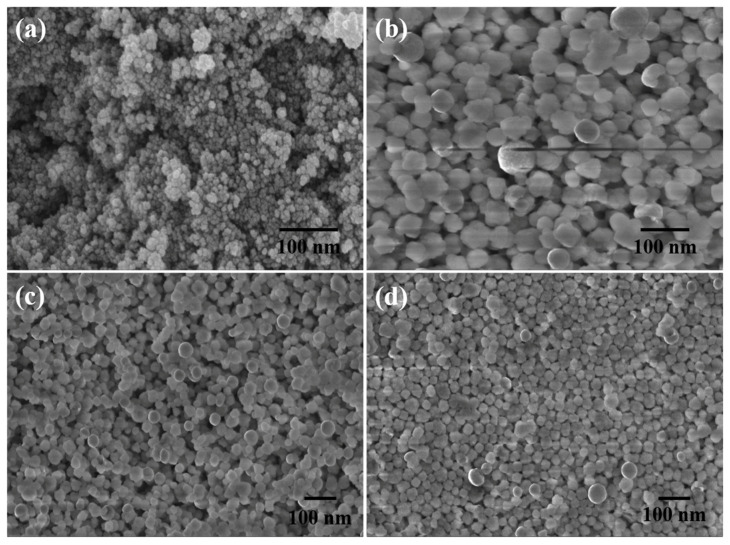
FESEM scans of (**a**) Fe_3_O_4_ nanoparticles, (**b**) IRMOF-3, (**c**) Fe3O4@IRMOF-3, and (**d**) Fe_3_O_4_@IRMOF-3/FA [118]. Reprinted with permission from Ref. [118]. Copyright 2016 Royal Society of Chemistry.

**Figure 5 molecules-27-00100-f005:**
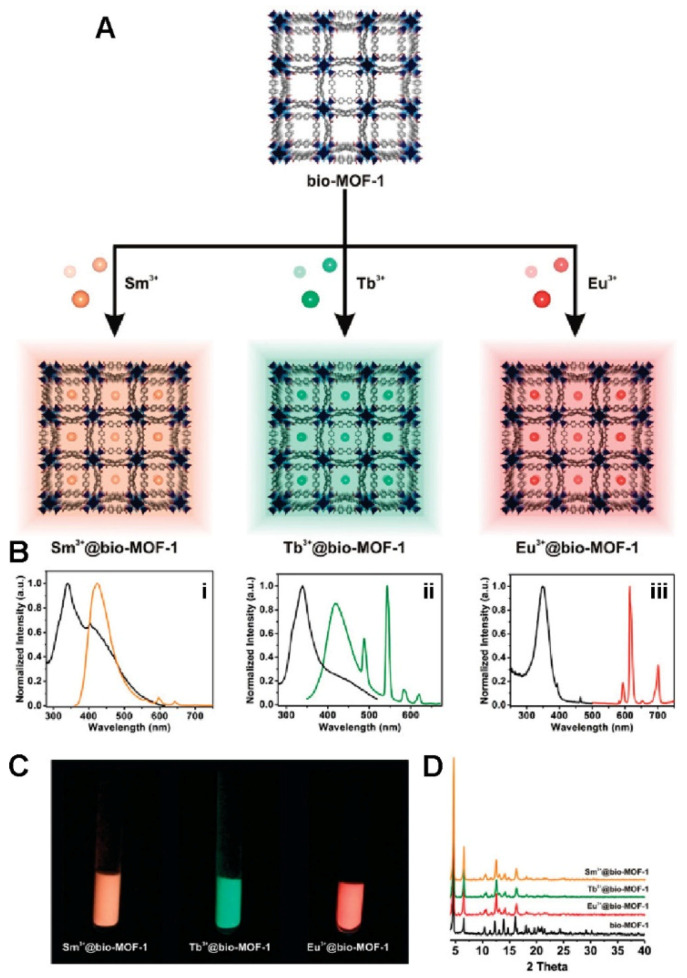
(**A**) Bio-MOF-1 encapsulation and sensing of lanthanide cations and (**B**) Bio-MOF-1 with transition metals cations: absorption and emission spectra of Sm^3+^@bio-MOF-1 (i), Tb^3+^@bio-MOF-1 (ii), and Eu^3+^@bio-MOF-1 (iii). (**C**) Samples of Ln^3+^@bio-MOF-1 with transition metals cations illu-minated with 365 nm UV light. (**D**) XRPD patterns of Ln^3+^@bio-MOF-1. Reprinted with permission from Ref. [125]. Copyright 2011 American Chemical Society.

**Figure 6 molecules-27-00100-f006:**
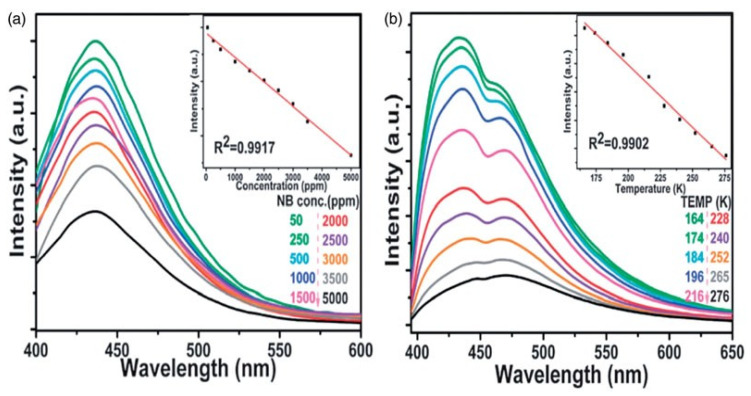
(**a**) Zn–TDPAT emission spectra (kex at lower wavelength) at varied concentration levels and (**b**) various thermal values [131]. Reprinted with the permission of Ref. [126]. Copyright 2013 Royal Society of Chemistry.

**Table 1 molecules-27-00100-t001:** Details about recently developed Zn MOFs: their structure, drug loading and release rate.

Zn-Based MOF		bio-MOF-1	IFMC-1	Med-MOF-1	[Zn_2_(1,4-bdc)_2_(dabco)n]	Zn-TBDA	MOF-74
Chemical/Empirical Formula		Zn_8_ (Ad)_4_ (BPDC)_6_ O_2_ (NH_2_ (CH_3_)_2_) + 8DMF, 11H_2_O	-	Zn_3_ (curcumin)_27_(DMA)_3_ (ethanol)	[Zn_2_ (1,4-bdc)_2_ (dabco)n]	[Zn (tbda)]_n_	Zn_2_DOT
Organic Linker		Adenatite	Triazole	Curcumin	1,4-diazabicyclo [2.2.2] octane (DBCO)	4′-(1H-tetrazol-5-yl)- [1,1′-biphenyl]-3,5-dicarboxylic acid	2,5-di hydroxyterephthalic acid
Drug		Procainamide	5-Fluorouracil	Ibuprofen	Ibuprofen	Methotrexate	Ibuprofen
Loading Degree		0.22 *g*/*g*	30.48 wt%	0.24 *g*/*g*	15 wt%	12.59%	313 k
Release Rate	%	20	89.8	97	80	61	-
	Time	72 h	120 h	80 h	288 h	48 h	-
Reference		[80]	[81]	[82]	[83]	[84]	[85]

**Table 2 molecules-27-00100-t002:** Stability and toxicity associated with various Zn-based metal organic frameworks.

Zn-MOFs	Stability Studies	Toxicity		Mechanism of Toxicity	Reference
		Cell Lines	IC50		
nanoZIF-8	-	HeLA J774	436 mm 109 mm	Breakdown of frameork into its constituents in the cell and endosomal environment	[147]
IRMOF-3	Stable at a temperature of 450 °C in the presence of N_2_ gas	PC12	Negligible at (25 g/mL) Considerable at (100 g/mL)	Disruption of cellular zinc homeostasis and down-regulation of GAP-43 protein	[92]
Nano ZIF-8	Stable at a temperature of 55 °C in the presence of N_2_ gas and in PBS (pH 7.4) for 7 days Stable in water	NCI-H292 HT-29 HL-60	>25 mg/mL	Less cytotoxicity of ZIF-8 is linked with the gentle release of drug	[146]
CS/Bio-MOF	Stable in PBS	MCF-7	3.1251 g/mL	-	[143]
ZIF-7	Stable in fetal bovine serum (10%)	MCF-7	Moderate toxicity	Slow release of drug	[148]
Zn-MOF-74	Stable in fetal bovine serum (10%)	HepG2 MCF7	High Toxicity	Viability 38.8 ± 3.6% at 200 mM Viability 57.6 ± 0.6% at 200 mM	[149]

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
